# Single Nucleotide Polymorphisms as Biomarkers of Response to Neoadjuvant Chemoradiotherapy in Rectal Cancer: A Systematic Review

**DOI:** 10.3390/cancers17243995

**Published:** 2025-12-15

**Authors:** Katarzyna Połomska, Magda Rybicka, Adrianna Jażdżewska, Magdalena Prud, Stefania Jackowska, Jaroslaw Kobiela, Piotr Spychalski

**Affiliations:** 1Department of Surgical Oncology, Transplant Surgery and General Surgery, Medical University of Gdansk, Sklodowska 3A Str., 80-210 Gdansk, Polandadrianna.jazdzewska@gumed.edu.pl (A.J.); kobiela@gumed.edu.pl (J.K.); 2Intercollegiate Faculty of Biotechnology, University of Gdansk and Medical University of Gdansk, Abrahama 58, 80-307 Gdansk, Poland; magda.rybicka@biotech.ug.edu.pl (M.R.);

**Keywords:** rectal cancer, single nucleotide polymorphism, neoadjuvant chemoradiotherapy, complete pathological response

## Abstract

Predicting the response of patients with rectal cancer to preoperative chemoradiotherapy is a major challenge in oncology. It has been hypothesized that variations in genes, called single nucleotide polymorphisms (SNPs), could explain why some patients achieve complete tumor regression while others do not. We systematically reviewed published studies that analyzed SNPs as possible markers of treatment response. We found no single genetic marker that consistently predicts how tumors respond to therapy. These results show that the current evidence is too weak to guide treatment decisions based on SNPs. Future research should focus on large, well-designed studies combining genetics with other biological and clinical data to build reliable prediction models.

## 1. Introduction

According to the latest NCCN Guidelines, neoadjuvant therapy is standard for patients with locally advanced rectal cancer (stage II or III, T3–T4 or N+), typically involving neoadjuvant chemoradiotherapy (nCRT) or total neoadjuvant therapy (TNT), and is especially recommended for tumors with high-risk features such as threatened mesorectal fascia or low tumor location [1]. The recommended approach is either long-course chemoradiotherapy using fluoropyrimidine-based chemotherapy (5-fluorouracil or capecitabine) administered concurrently with pelvic radiation, or total neoadjuvant therapy (TNT), which involves the administration of systemic chemotherapy (commonly FOLFOX or CAPOX) and radiation before surgery. TNT can be administered as induction chemotherapy followed by CRT, or as CRT followed by consolidation chemotherapy, and is especially recommended for high-risk features.

A key indicator of efficacy is the achievement of a pathological complete response (pCR), defined as the absence of residual tumor cells in the resected specimen after neoadjuvant treatment. In clinical practice, the proportion of patients who achieve pCR varies between 15% and 30%, depending on initial tumor burden and the specific treatment regimen. The PRODIGE-23 [2] and RAPIDO [3] trials reported pCR rates of 27.8% and 27.4%, respectively. Thus, one quarter of patients undergoing surgery (resection or abdominoperineal resection) after neoadjuvant therapy were found to have no detectable tumor cells on histopathological examination of the operative specimen. This has led to increasing interest in non-operative management strategies such as the “watch-and-wait” approach [1].

However, the ability to accurately predict which patients will respond favorably to nCRT remains limited. Single nucleotide polymorphisms (SNPs), as stable and accessible genetic markers, have shown potential in predicting radiosensitivity and chemosensitivity by influencing DNA repair capacity, cell cycle regulation, and drug metabolism. Systematic analyses of previously published studies have been constrained by the limited number of articles published on this topic and their heterogeneity [4,5,6,7,8]. This systematic review aims to synthesize the current evidence on SNPs associated with the response to nCRT in rectal cancer. This may aid in identifying genetic predictors that could stratify patients by their likelihood of response, thereby supporting personalized treatment decisions and potentially sparing selected patients from unnecessary surgery and its associated morbidity.

This study aims to systematically evaluate and synthesize the evidence on the associations between specific SNPs and the response to nCRT in patients with rectal cancer.

## 2. Materials and Methods

### 2.1. Search Strategy

The systematic review was based on the patients, interventions, comparisons, outcomes (PICO) framework and was conducted according to the Preferred Reporting Items for Systematic Reviews and Meta-Analyses (PRISMA) statement [9]. A search of the PubMed and Web of Science databases was performed to identify relevant articles without the use of filters. Backward citation chaining of the reference lists of all eligible full-text articles was performed to identify additional studies. This study was registered with PROSPERO (Identifier: CRD420251054562).

### 2.2. Evidence Acquisition

Databases were searched on 17 May 2025 to identify relevant articles. The search was conducted without using filters. We used a combination of keywords and Medical Subject Headings (MeSH) terms for the search. The following query was created using Boolean search terms: (SNP OR polymorphism) AND (colorectal cancer OR rectal cancer) AND (chemotherapy OR radiotherapy OR neoadjuvant).

The screening protocol is shown in the PRISMA flowchart (Figure 1). The initial search yielded 1976 results.

Machine-learning–assisted screening tool was applied for record selection to reduce screening time and improve accuracy [10]. Records were deduplicated with the DistillerSR (Evidence Partners Inc., Ottawa, ON, Canada) web application [11], and 1424 records were further processed.

Screening was further performed using DistillerSR [11]. The algorithm was initially trained with five relevant and five irrelevant studies and was iteratively retrained based on new included and excluded studies. A total of 558 studies were manually labeled by two independent authors (KP and SJ). We adopted a stopping criterion of 150 subsequent irrelevant articles since the last relevant (van Dijk et al. proposed a criterion of 100 irrelevant articles since last relevant) [12]. We identified 73 articles for full-text screening.

Full-text screening was performed manually by two independent researchers (KP and AJ) to identify the eligible studies. Disagreements between the researchers were resolved through discussion and consultation with another author (PS). After full-text screening, 32 articles were included in this review.

The following data were retrieved from the included studies: first author, title, journal, year of publication, study location (institution and country), study design, recruitment period, and sample size. We collected detailed data on patient characteristics, including age, sex, clinical staging, and type of surgery (if performed). For each study, we extracted information on the specific single nucleotide polymorphisms (SNPs) evaluated, methods of SNP detection, and the outcomes assessed pCR and tumor regression grade (TRG). Additionally, we recorded the definitions of response criteria, statistical methods used, effect sizes (e.g., odds ratios, hazard ratios, confidence intervals), and whether statistically significant associations were found. For evidence synthesis, odds ratios were directionally harmonized by inverting estimates where necessary so that all effect sizes consistently reflected the association with favorable treatment response. Data were independently extracted by two reviewers using a standardized form.

### 2.3. PICO Framework: Inclusion and Exclusion Criteria

We included original studies involving adult patients (≥18 years) with locally advanced rectal cancer treated with nCRT or neoadjuvant radiotherapy and surgery. The eligibility criteria included studies that reported the presence of single nucleotide polymorphisms (SNPs) and their association with treatment response, defined as pathological complete response (pCR) or tumor regression grade, and studies that provided a comparison between different SNP genotypes (wild-type, heterozygous, and homozygous). Studies focusing on other types of genetic alterations (e.g., MSI, gene expression) without SNP analysis and non-original reports (e.g., reviews, case reports, abstracts) were excluded. The PICO-based strategy adopted to identify relevant studies is presented in Table 1. Studies published in languages other than English were excluded from the review. No time constraints were imposed.

### 2.4. Evidence Synthesis and Risk of Bias Assessment

The QGenie tool was used to assess the risk of bias in the included studies. The risk of bias was evaluated by two independent authors (KP, AJ), and any disagreement was resolved via discussion and consensus [13].

## 3. Results

After screening and full-text assessment (73 articles), 32 articles were included in this review. All studies were genetic association studies; 12 were retrospective cohorts, and 20 were prospective cohorts (3 studies were pooled cohorts from randomized clinical trials). A total of 304 SNPs in 126 genes were analyzed. The total number of patients was 4116, with individual sample sizes ranging from 21 to 316. Most patients had stage II or III rectal cancer. Across the included studies, the neoadjuvant treatment protocols demonstrated substantial heterogeneity in both radiotherapy (RT) dose and concurrent chemotherapy regimens. Most studies used long-course chemoradiotherapy with total RT doses ranging from 45 to 50.4 Gy, delivered in 1.8–2.0 Gy fractions, frequently with a boost dose of 5.4–10 Gy. Fluoropyrimidine-based chemotherapy was the most widely used approach, with regimens consisting of continuous infusion of 5-fluorouracil (5-FU), capecitabine, tegafur/uracil (UFT), leucovorin, or combinations thereof. Several protocols included oxaliplatin-based doublets such as XELOX, mFOLFOX6, or CAPOX (with or without cetuximab). Less commonly, studies adopted intensified RT schedules (e.g., 60–65 Gy) or combined external-beam RT with brachytherapy. Furthermore, several studies administered more than one treatment protocol to the same cohort.

Across the included studies, 304 SNPs in 126 genes were analyzed. The number of SNPs analyzed per study varied widely: most studies assessed a small panel (1–10 SNPs), often targeting SNPs with prior pharmacogenetic relevance rather than using genome-wide strategies; only one study employed genome-wide screening of over 690,000 SNPs [14,15].

The most frequent genotyping methods were PCR-RFLP (eight studies), TaqMan (five studies), Sanger sequencing (four studies), and SNaPshot (three studies). Biological material for genotyping was derived from peripheral blood (21 studies), pre-treatment tumor tissue (8 studies), tumor samples (4 studies), and non-tumor tissue samples (1 study). The investigated outcomes were pathological complete response (pCR) in 16 studies and TRG in 18 studies. TRG was assessed using different scoring systems (Mandard, Dworak, and AJCC). The proportion of patients achieving pCR ranged from 11% to 51% across studies, and the proportion of responders to neoadjuvant treatment ranged from 19% to 84%. Across studies using the Mandard TRG system, responder definitions varied: eight studies defined responders as Mandard TRG 1–2, one study as Mandard TRG 1, and one study as Mandard TRG 1–3. Heterogeneity was also observed among studies employing the Dworak TRG system, where responders were defined as Dworak TRG 2–4 in two studies, modified Dworak TRG 2–3 in one study, and Dworak TRG 3–4 in three studies. Of the 16 studies assessing pCR, 12 defined pCR as the absence of residual tumor cells. One study specified pCR as no evidence of residual carcinoma or only residual microfoci; two studies (on the same cohort) allowed radiological assessment of complete response in patients who did not undergo surgery; and one study did not provide a definition of pCR. The detailed baseline characteristics of the included studies are presented in Table 2.

Genes involved in DNA repair pathways were the most frequently studied, particularly *XRCC1* (14 SNPs analyzed in 11 studies), *XRCC3* (10 SNPs in 5 studies), *ERCC1* (7 SNPs in 9 studies), and *ERCC2* (9 SNPs in 7 studies). The second most common group was folate metabolism genes, particularly *MTHFR* (11 SNPs in 12 studies) and *TYMS* (11 SNPs in 7 studies). The other most frequently studied genes were *EGFR* (7 SNPs in 7 studies) and *GSTP1* (7 SNPs in 6 studies). Other frequently analyzed SNPs were located in genes related to immune response, miRNAs, reactive oxygen species, angiogenesis, and cell cycle regulation (Table 3).

Of the 304 SNPs analyzed, only 2 SNPs were associated with a pathological response in more than one study (rs25487 *XRCC1* and rs1801133 *MTHFR*); most (22 SNPs) associations were reported in single studies only (Table 4). For each SNP assessed in multiple studies, the cumulative number of patients from studies that did not demonstrate a significant association with pathological response greatly exceeded the number of patients from studies that did. Even for the most widely studied SNPs (*XRCC1* rs25487 and *MTHFR* rs1801133), the number of patients in negative studies significantly outweighed the number of patients in positive studies. The list of SNPs that were not associated with pathological response in any of the included studies is presented in Appendix A.

### 3.1. rs25487 (XRCC1)

This SNP was analyzed in 10 studies, of which 4 demonstrated its association with pathological response after nCRT. Studies analyzing rs25487 varied in terms of outcome measures, nCRT schemes, and genotyping methods. Furthermore, studies have identified different genotypes related to pathological responses. Balboa et al. [15] found that the AA genotype was significantly associated with improved treatment response after adjustment for sex, age, and cancer stage (odds ratio [OR] 7.93; 95% CI: 1.03–60.83; *p* = 0.036). In the cohort analyzed by Grimminger et al., the AG genotype was linked to a higher major response rate (47%) than the AA or GG genotypes (22%; *p* = 0.039). Conversely, Formica et al. reported a strong association between the GG genotype and better tumor regression (OR 25.8; *p* = 0.049), and Lamas et al. identified the GG genotype as predictive of a favorable response over the GA genotype (OR 4.18; 95% CI: 1.62–10.74; *p* = 0.003).

### 3.2. rs1801133 (MTHFR)

This SNP was analyzed in 10 studies, of which three demonstrated its association with pathological response after nCRT. Multivariable analysis by Cecchin et al. found that patients carrying at least one T allele (CT or TT genotypes) had a significantly lower likelihood of achieving tumor regression (TRG ≤ 2) than those with the CC genotype (OR = 0.48; 95% CI: 0.24–0.96; *p* = 0.034). Nikas et al. reported that the CC genotype was associated with a higher probability of pathological response relative to the CT and TT genotypes (OR = 2.91; 95% CI: 1.23–6.89; *p* = 0.015). Similarly, Terrazzino et al. demonstrated a greater response rate among CC homozygotes than among T allele carriers (responders: 57% vs. 34%; OR = 0.32; 95% CI: 0.14–0.71; *p* < 0.006).

### 3.3. Risk of Bias

The risk of bias analysis is presented in Table 5. Overall quality scores ranged from 34 to 65 out of a maximum possible score of 77, with a mean of 51.72 (SD = 6.79) and a median of 52. Only two studies (6.2%) achieved good quality ratings (≥60 points), while the majority (56.2%, *n* = 18) were classified as moderate quality (50–59 points), and over one-third (37.5%, n = 12) demonstrated poor methodological quality (<50 points). Several critical methodological domains consistently scored below acceptable thresholds across the included studies. The sample size and statistical power considerations were particularly weak, averaging only 3.47 out of 7 points, indicating that most studies were underpowered and lacked formal sample size calculations. The assessment and control of other sources of bias were similarly inadequate (mean score = 3.72). Four studies exhibited particularly severe methodological limitations with overall scores below 45: Spindler 2006 (score 34) [41], Stoehlmacher 2008 (score 38) [43], Xiao 2016 (score 41) [45], and Rampazzo 2020 (score 43) [37].

## 4. Discussion

This systematic review synthesizes evidence from 32 studies, including 4116 patients and evaluating 304 SNPs across 126 genes, assessing their association with the response to neoadjuvant chemoradiotherapy in patients with locally advanced rectal cancer. Despite the growing body of literature on this subject, no polymorphism has emerged as a reliable or consistently reproducible predictor of treatment response. Overall, the findings were largely inconsistent and non-reproducible. Our findings are consistent with recent meta-analyses and systematic reviews, which also report a lack of robust or reproducible associations between individual polymorphisms and response to nCRT [8,46,47]. These results collectively emphasize the need for validation in larger, independent cohorts and suggest that current candidate gene approaches may be insufficient. The most frequently studied polymorphisms were located within genes involved in DNA damage repair pathways, particularly *XRCC1*, *XRCC3*, *ERCC1*, and *ERCC2*, and in genes related to folate metabolism, such as *MTHFR* and *TYMS*. *XRCC1*, *XRCC3*, *ERCC1*, and *ERCC2* encode proteins crucial for the repair of DNA damage induced by ionizing radiation, directly affecting tumor cell radiosensitivity. *MTHFR* and *TYMS* play key roles in folate metabolism and nucleotide synthesis, influencing the effectiveness of fluoropyrimidine-based chemotherapy. In addition, variants within genes related to oxidative stress, immune regulation, and cell cycle control, such as GSTP1, EGF, IL13, FPR1, and TERT, were reported to show favorable associations with pathological complete response or tumor regression grade in single studies, but these signals were rarely replicated and generally remained exploratory.

Most studies adopted a candidate-driven approach, focusing on SNPs in these pathways. However, most SNPs were assessed in isolated studies, with the most frequently reported associations involving rs25487 (*XRCC1*) and rs1801133 (*MTHFR*). Even for these commonly investigated SNPs, the results remain highly inconsistent. For example, different cohorts identified either the wild-type or the homozygous variant genotype of *MTHFR* rs1801133 as favorable, while other studies did not observe any significant association, which substantially weakens the biological plausibility of a true effect and points to methodological variability.

For all SNPs analyzed in multiple studies, reports indicating no significant association with pathological response outnumber those demonstrating any positive correlation. This inconsistency likely reflects considerable heterogeneity among the included studies regarding outcome definitions (pCR, response according to different TRG systems with varying response cut-offs), nCRT protocols (variable chemotherapy regimens and radiotherapy doses), sample sizes, and analytical models (dominant, recessive, and additive models). In addition, the included studies used a wide range of genotyping platforms, ranging from low-throughput, locus-specific assays such as PCR-RFLP and TaqMan to sequencing- or array-based approaches (Sanger sequencing, SNP arrays, SNaPshot, and MassARRAY), each characterized by distinct analytical sensitivities, multiplexing capacities, and susceptibility to technical artifacts, which may have introduced further between-study variability in SNP calls and effect estimates. Detailed quality control metrics for these assays, including reproducibility, call rates, and the systematic use of positive and negative controls, have only been sparsely reported or are entirely missing in many studies, limiting a rigorous appraisal of genotyping reliability. Nearly 90% of the included studies analyzed SNPs in Caucasian patients, while the remaining studies included Asian patients; other ethnic groups were not represented. This limits the generalizability of the findings due to potential population-specific effects, such as differences in allele frequency and linkage disequilibrium (LD) structure. It is important to note significant heterogeneity in cancer staging across the analyzed studies, particularly the varying proportions of patients with stage II rectal cancer according to the AJCC. Patients with stage II cancer tend to have a better response to neoadjuvant treatment, which could facilitate an analysis of the association of SNP and complete pathological response in some studies.

Another hypothesis explaining the non-reproducibility of the results is the type I error (false-positive findings). Of all SNP analyses conducted (with each SNP analysis in each publication treated as a separate test), positive results were found in 30 of 407 separate statistical analyses (approximately 7%). Considering the plausible publication bias, where positive results are preferentially published, the true proportion of significant findings is likely to be lower, probably below 5%, which aligns with the nominal Type I error rate. Given the relatively small and often underpowered cohorts and the large number of statistical tests performed without systematic correction for multiple comparisons, the small proportion of positive results is close to what would be expected by chance alone, particularly in the presence of publication bias favoring significant findings.

This aligns with important limitations emerging consistently across the current body of literature, including generally small sample sizes with no reported power calculations, insufficient adjustment for multiple comparisons, substantial variability in treatment protocols and definitions of treatment response, limited use of multivariable analytical models, and considerable population heterogeneity with minimal consideration of environmental and epigenetic influences on the results. A risk of bias assessment using QGenie, a tool specifically developed for genetic association studies, highlighted suboptimal reporting and study design in several publications. This is particularly evident in the justification of sample sizes, handling of confounders, and replication strategies. Moreover, five studies with the lowest risk of bias (highest QGenie score) reported no significant SNP associated with the response to nCRT.

Our data highlight the ongoing uncertainty regarding the clinical utility of SNPs in predicting the nCRT response in rectal cancer. No individual variant has been sufficiently validated for clinical implementation. Importantly, none of the available genetic markers have shown a superior predictive value over the current clinicopathological criteria. Consequently, treatment decisions for patients with locally advanced rectal cancer should continue to be based on established clinical and pathological factors, radiological staging, and patient preference. Germline SNP testing for predicting the response to nCRT cannot currently be recommended outside of research settings. Conversely, somatic mutations have been identified to play a role in predicting the response to nCRT. KRAS mutation is independently associated with a lower pCR rate in locally advanced rectal cancer [48,49]. Microsatellite instability is independently associated with a reduction in pCR for locally advanced rectal cancer after nCRT [50]. This is reflected in current guidelines identifying checkpoint inhibitors as the preferred initial therapy for patients with microsatellite instability [1].

Future progress in the field may depend on shifting from single SNP analyses toward polygenic risk models and the combination of genetic data with other-omic layers (e.g., transcriptomics and proteomics). The application of machine learning, coupled with standardized nCRT protocols and outcome definitions, could help improve reproducibility and predictive accuracy. Genome-wide or large-panel approaches, coupled with rigorous replication and external validation and integrated with tumor-intrinsic features (somatic alterations, gene expression, epigenetic markers), circulating biomarkers, and advanced imaging, may enable the development of multi-omic prediction tools that better capture the complex biology of chemoradiotherapy response.

Before any genetic biomarker can be applied in clinical practice, several barriers must be addressed: robust validation, demonstration of added predictive value, cost-effectiveness, and accessibility of testing. Large-scale prospective, multicenter studies with standardized methodologies, sufficient sample sizes, multivariable analyses, and appropriate statistical corrections are urgently needed to clarify the true potential of genetic variants in guiding personalized therapy for rectal cancer.

## 5. Conclusions

This systematic review, covering 32 studies and 4116 patients, indicates that no individual germline SNP has shown a consistent association with the response to nCRT in locally advanced rectal cancer. Although some variants in DNA repair, folate metabolism, and other pathways have been highlighted as potential predictors in isolated analyses, their effects have not been reproduced across independent cohorts or under uniform response definitions. At present, the evidence base does not justify using individual SNPs as standalone biomarkers to guide neoadjuvant treatment decisions in rectal cancer. Risk stratification and selection of candidates for organ-preserving strategies, including watch-and-wait, should therefore continue to rely on established clinicopathological factors together with high-quality imaging and standardized response assessment. Future work should focus on sufficiently powered, multicenter prospective studies using harmonized neoadjuvant treatment protocols and uniform response criteria, encompassing both pathological complete response and validated tumor regression grading systems. In this setting, polygenic models and integrative multi-omic strategies that combine germline variation with tumor molecular profiles, imaging-derived features, and key clinical variables are more likely than single markers to yield clinically useful tools for predicting chemoradiotherapy response in rectal cancer.

## Figures and Tables

**Figure 1 cancers-17-03995-f001:**
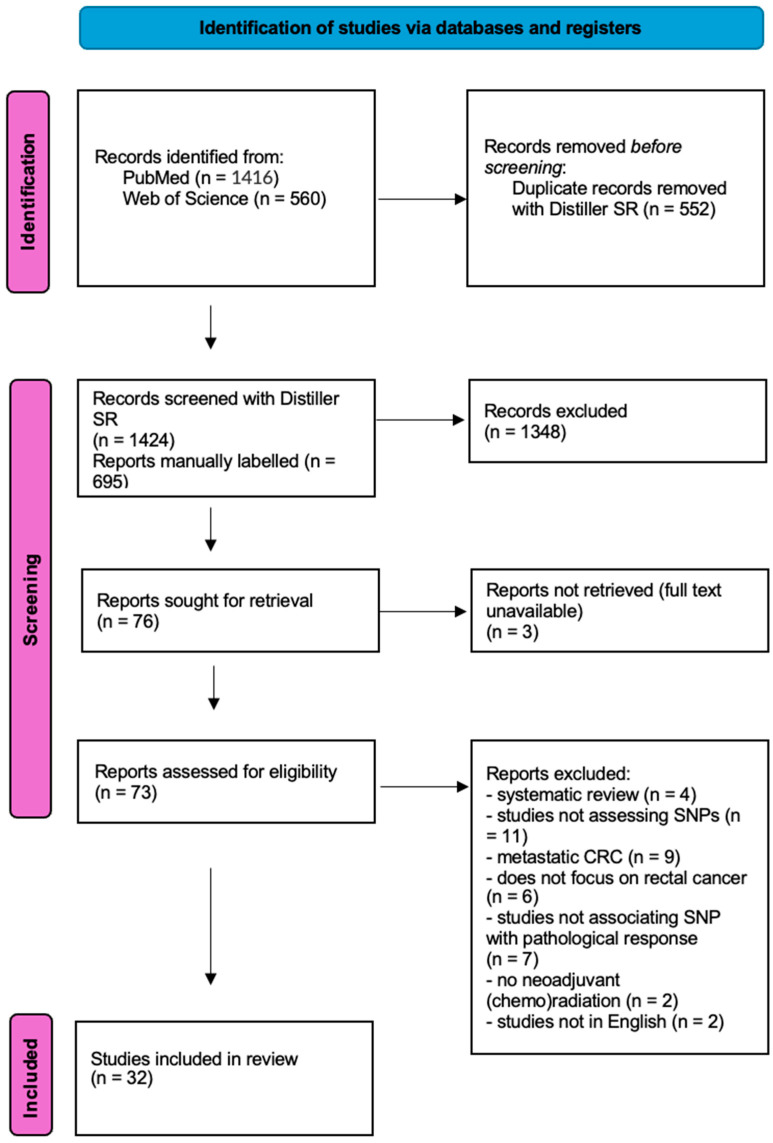
PRISMA screening protocol.

**Table 1 cancers-17-03995-t001:** PICO framework for the systematic review.

	Description
P—Population	Adult patients (≥18 years) diagnosed with locally advanced non-metastatic rectal cancer who underwent neoadjuvant (chemo)radiotherapy (nCRT).
E—Exposure	Presence of specific single nucleotide polymorphisms (SNPs).
C—Comparator	Patients with alternative SNP variants or wild-type genotypes.
O—Outcomes	Pathological complete response (pCR) or Tumor regression grade (TRG)

**Table 2 cancers-17-03995-t002:** Baseline characteristics of the included studies.

Author	Year	Study Design	Country	Number of Patients	Years of Recruitment	Initial Staging	Neoadjuvant Protocol	Median Age	Sample Examined	Number of SNPs Analyzed	Genes Investigated	Genotyping Method	Primary Outcome	Responder/pCR Definition and Rate	Positive Predictors of Tumor Response
Balboa [15]	2010	Retrospective cohort	Spain	65	NR	stage II (31%), stage III (69%)	RT + Uracil/Capecitabine	64	peripheral blood sample, pre-treatment tumor material	7	*XRCC1*, *ERCC1*, *ERCC2* (*XPD*), *MTHFR*, *DPYD*, *TYMS*, *EGFR*	SNaPshot	Mandard TRG	responders (TRG 1–2) 48% non-responders (TRG 3–5) 53%	*XRCC1* - rs25487 AA
Boige [16]	2019	Prospective cohort (RCT subgroup)	France	316	2005–2008	T3–4 M0 (100%)	RT + Capecitabine (49%) or dose-intensified RT + Capecitabine + Oxaliplatin (51%)	61	peripheral blood sample	66	*ERCC1*, *ERCC2* (*XPD*), *ERCC4*, *XRCC1*, *XRCC3*, *XPA*, *GSTP1*, *MTHFR*, *TYMS*, *SOD2*	Illumina Infinium iSelect custom SNP genotyping	Dworak TRG (modified)	responders (TRG 2–3) 37.6% non-responders (TRG 0–1) 62.4%	*ERCC2* - rs1799787 C>T *ERCC1* - rs10412761 A>G
Cecchin [17]	2011	Retrospective cohort	Italy	238	1993–2006	staging not specified	RT 45–50.4 Gy + 5-FU ± others	61	peripheral blood sample	21	*MTHFR*, *ABCB1*, *ABCC2*, *GSTA1*B*, *RAD51*, *MLH1*, *MSH2*, *OGG1*, *XRCC3*, *XRCC1*, *ERCC2* (*XPD*), *ERCC1*, *GSTP1*	Pyrosequencing, TaqMan, Gel electrophoresis	Mandard TRG	good responders (TRG 1–2) 51% intermediate responders (TRG 3) 21% non-responders (TRG 4–5) 27%	*hOGG1* - rs1052133 CC *MTHFR* - rs1801133 TT
Chiang [18]	2021	Retrospective cohort + animal model	Taiwan	130	2006–2014	cT3–4 or cN+ (100%)	RT 50.4 Gy + 5-FU/UFT/Capecitabine	59.7	non-tumor surgical material	4	*FPR1*, *TIM3*, *P2RX7*, *TLR1*	MassARRAY	Dworak TRG	responders (TRG 3–4) 65% non-responders (TRG 1–2) 35%	*FPR1* - rs867228 AC/AA
Dreussi [19]	2016	Retrospective cohort	Italy	280	1993–2011	T3–T4 and N0–2 M0 (100%)	RT 50.4 Gy/55.0 Gy + 5-FU/Capecitabine ± Oxaliplatin	61	peripheral blood sample	30	*MTHFR*, *MSH6*, *XRCC1*, *OGG1*, *MDM2*, *MLH1*, *MGMT*, *GSTP1*, *SOD2*, *XRCC3*, *TP53*, *ATM*, *EGFR*, *PARP-1*, *EXO1*, *ERCC1*, *ERCC2* (*XPD*), *EGF*, *VEGF*, *APEX1*	TaqMan PCR, Pyrosequencing, Gel electrophoresis	pCR	pCR (Mandard TRG 1) 28%	none
Dreussi [20]	2016	Prospective cohort	Italy	265	1993–2011	T3–T4 N0–2 M0 (100%)	RT 50.4/55.0 Gy + 5-FU/Capecitabine ± Oxaliplatin	NR	peripheral blood sample	114	*SMAD3*, *TRBP*, *DROSHA*, *CNOT4*, *CNOT6*, *DDX20*, *DGCR8*, *DICER1*, *SMAD2*, *SMAD5*, *TNRC6A*, *TNRC6B*, *miR-196a-2*, *miR-371A* (authors do not report all analyzed SNPs)	BeadXpress platform	pCR	pCR (Mandard TRG 1) 28%	*SMAD3* - rs744910 AG/GG - rs745103 GG - rs17228212 TT *TRBP* - rs6088619 AG/GG *DROSHA* - rs10719 CC
Dzhugashvili [21]	2014	Retrospective cohort	Spain	159	2004–2010	cT0–2 (5.7%) cT3–4 (94.3%) cN+ (72.9%)	RT 50.4 Gy (1.8 Gy per fraction) + Capecitabine	64	peripheral blood sample	6	*IL1B*, *PTGS1*, *PTGS2*	TaqMan	pCR	pCR (ypT0N0) 18.2%	none
Formica [22]	2018	Prospective cohort	Italy	51	NR	stage II (35%), stage III (65%)	RT 45 Gy (1.8 Gy per fraction) + 5.4 Gy boost + Cisplatin + Capecitabine	63	surgical tumor material	5	*GSTP1*, *XRCC1*, *ERCC1*, *MTHFR*, *ABCB1*	Pyrosequencing	AJCC TRG	responders (AJCC TRG 0–1) 34% non-responders (AJCC TRG 2–3) 66%	*XRCC1* - rs25487 GG
Garcia-Aguilar [23]	2011	Prospective cohort (Secondary analysis of phase II trial)	USA, Spain	132	2004–2012	stage II (28%), stage III (69%), unknown (3%)	RT 50.4 Gy, + 5-FU ± mFOLFOX-6	57	pre-treatment tumor material	2	*CCND1*, *MTHFR*	Sanger sequencing	pCR	pCR (AJCC TRG 0) 25%	*CCND1* - rs603965 AG/GG *MTHFR* - rs1801133 TC/CC
Grimminger [24]	2010	Prospective cohort	Germany	81	1997–2008	T3/4 Nx (100%)	RT 50.4 Gy + 5-FU	59	pre-treatment tumor material	3	*XRCC1*	TaqMan	Viable residual tumor cells	major response (VRTC 3–4) 32% minor response (VRTC 1–2) 68%	*XRCC1* - rs25487 AG
Havelund [25]	2012	Prospective cohort	Denmark	198	2005–2009	cT3 (83%), cT4 (17%); N+ (89%)	RT 50.4 Gy with or ± 10 Gy brachytherapy + UFT + Leucovorin	63	peripheral blood sample	3	*HIF-1α*	KASPar, Sanger sequencing, TaqMan	Mandard TRG	responders (TRG 1) 19% non-responders (TRG 2–4) 81%	none
Ho-Pun-Cheung [26]	2011	Prospective study	France	71	2005–2008	stage II (20%), stage III (72%), stage IV (9%)	RT 45/50 Gy (1.8–2 Gy per fraction) + Capecitabine ± Oxaliplatin	61	peripheral blood sample	128	*ADPRT*, *CHEK1*, *CHEK2*, *ATM*, *BRCA1*, *BRCA2*, *ERCC1*, *ERCC2* (*XPD*), *ERCC4*, *ERCC5*, *LIG3*, *LIG4*, *MBD4*, *MGMT*, *XPA*, *XRCC1*, *XRCC3*, *BAX*, *CASP3*, *CASP8*, *CASP10*, *CASP9*, *BCL2*, *CCND1*, *CDKN1A*, *MDM2*, *TP53*, *TP73*, *EGF*, *EGFR*, *ERBB2*, *IGF1*, *FGF2*, *FGFR4*, *IGF2R*, *TGFB1*, *VEGF*, *FCGR2A*, *FCGR3A*, *IL8*, *IL10*, *IL1B*, *IL4*, *IL6*, *IL13*, *LTA*, *NFKB1*, *TNFA*, *PTGS2*, *PPARG*, *NFE2L2*, *PARP-1*, *MPO*, *GPX1*, *SOD2*, *NOS2A*, *NOS3*, *HIF-1A*, *CYP1A1*, *GSTP1*, *GSTT1*, *MT-ND3*, *MTHFR*, *ICAM5*, *GSK3B*, *CTNNB1*, *APEX1*, *NBN*, *RECQL*, *ZNF350*, *RAD52*, *XRCC5*	SNPlex Genotyping System, TaqMan, PCR-RFLP	Dworak TRG	responders (TRG 3–4) 45% non-responders (TRG 0–2) 55%	*SOD2* - rs4880 CC *IL13* - rs1800925 CC
Ho-Pun-Cheung [27]	2007	Prospective cohort	France	70	1996–2001	stage I (16%), stage II (31%), stage III (36%), stage IV (17%)	RT (45/60 Gy)	64	peripheral blood sample	1	*CCND1*	PCR-RFLP	Dworak TRG	responders (TRG 2–4) 50% non-responders (TRG 0–1) 43% NR 7%	*CCND1* - positive G870A AA
Hu-Lieskovan [28]	2011	Prospective cohort from Phase I/II trials	Germany, Slovenia, Belgium	130	NR	stage II (3%), stage III (84%), stage IV (12%)	RT + Cetuximab + Capecitabine/Oxaliplatin/5-FU	61	surgical tumor material	13	*EGFR*, *KRAS*, *IL8*, *MTHFR*, *FCGR2A*, *FCGR3A*, *XRCC3*, *VEGF*, *EGF*, *CCND1*, *PTGS2*, *RAD51*	PCR-RFLP	pCR	pCR (Dworak TRG 4) 15%	*EGF* - rs4444903 AG/GG
Hur [29]	2011	Prospective cohort	South Korea	44	2007–2008	T2(2.3%), T3 (40.9%), T4 (56.8%)	RT 45 Gy (1.8 Gy per fraction) + 5.4 Gy boost + 5-FU	58	pre-treatment tumor material	1	*TYMS*	Sanger sequencing	pCR, Mandard TRG	pCR (Mandard TRG 1) 14% responders (Mandard TRG 1–2) 41% non-responders (Mandard TRG 3–4) 59%	none
Kim [14]	2013	Prospective cohort	South Korea	113 (genome-wide screening of 691,162 SNPs)	NR	stage II (9%), stage III (89%), stage IV (2%)	RT 45 Gy + 5.4 Gy boost + 5-FU + Leucovorin (80%)/Capecitabine (20%)	59	peripheral blood sample	9	*FAM101A*, *CORO2A*, *USP20*, *ZNF281*, *OR2T4*, *SLC10A7*, *ASZ1*, *MED4*, *CDC42BPA*	Genome-Wide Human SNP Array, Pyrosequencing	Mandard TRG	responders (Mandard TRG 1–3) 84% non-responders (Mandard TRG 4) 16%	*CORO2A* - rs1985859 CC (wild type)
Kim [30]	2017	Prospective cohort	South Korea	91	2009–2012	T3 (97%), T4 (3%), N+ (87%)	RT 50.4 Gy (1.8 Gy per fraction) + Tegafur + Uracil + Leucovorin	59	peripheral blood sample	7	*UMPS*, *CYP2A6*, *ABCB1*	PCR-RFLP	pCR	pCR (not defined) 11%	none
Lamas [31]	2012	Prospective cohort	Spain	93	2007–2008	stage II (28%), stage III (72%)	RT 50.4 Gy + 5-FU	67	peripheral blood sample	5	*XRCC1*, *TYMS*, *MTHFR*, *ERCC1*	SNaPshot	Mandard TRG	responders (TRG 1–2) 47% non-responders (TRG 3–4) 53%	*XRCC1* - rs25487 GG/AA *TYMS* - 5′UTR VNTR 2R/3G, 3C/3G, 3G/3G
Leu [32]	2021	Prospective cohort	Germany	287	1998–2016	stage II (21%), stage III (79%)	RT 50.4 Gy (1.8 Gy per fraction) + 5-FU ± Oxaliplatin	64.4	peripheral blood sample	8	*SOD2*, *SOD3*, *CAT*, *CYBA*, *GPX1*, *MPO*, *OGG1*	SNaPshot	pCR	pCR (not defined) 17%	none
Nicosia [33]	2018	Retrospective cohort	Italy	80	2008–2015	T2 (11%) T3 (84%) T4 (5%), N+ (54%) M0 (100%)	RT 45 Gy + 10 Gy boost + Capecitabine (60%)/5-FU (40%)	64	peripheral blood sample	2	*GSTP1*, *XRCC1*	Pyrosequencing	pCR	pCR (Dworak TRG 4) 19%	*GSTP1* - rs1695 AA (wild type)
Nikas [34]	2015	Prospective cohort	USA	108	NR	staging not specified	RT 50.4 Gy + 5-FU	NR	peripheral blood sample	1	*MTHFR*	High-resolution Melting Analysis	pCR	pCR (College of American Pathologists TRG 0) 33% non-responders (College of American Pathologists TRG 3–4) 67%	*MTHFR* - rs1801133 CC (wild type)
Paez [35]	2011	Prospective cohort	Spain	128	1998–2009	T2 (6%), T3 (81%), T4 13%, N+ (60%)	RT 45 Gy+ 5-FU/Capecitabine/Capecitabine + Oxaliplatin/5-FU + Oxaliplatin	65	peripheral blood sample	10	*XRCC1*, *ERCC1*, *EGFR*, *GSTP1*, *ERCC2* (*XPD*), *TYMS*	TaqMan	pCR	responders (pCR Mandard TRG 1 plus residual microfoci) 43% non-responders 57%	none
Peng [36]	2018	Retrospective cohort	China	97	2008–2011	stage II (36.3%), stage III (63.7%)	RT 50 Gy (2 Gy per fraction) + XELOX	58	peripheral blood sample	12	*PTEN*, *PIK3CA*, *AKT1*, *AKT2*, *FRAP1*	PCR-RFLP	pCR, Dworak TRG	pCR (Dworak TRG 4) 14.4% responders (TRG 2–4) 69.1% non-responders (TRG 1) 30.9%	none
Rampazzo [37]	2020	Prospective cohort	Italy	194	NR	stage I (3.2%), stage II (11.1%), stage III (84.1%), stage IV (1.6%)	RT + Fluoropyrimidine ± other drug	65	peripheral blood sample	8	*TERT*	TaqMan	Mandard TRG	responders (TRG 1–2) 46% non-responders (TRG 3–5) 54%	*TERT* - rs2736108 CC - rs2853690 GG/AA
Sclafani [38]	2015	Retrospective cohort	UK, Spain, Sweden, and others	155	2005–2008	staging not specified	CAPOX (±Cetuximab) + Capecitabine-RT 45 Gy + 5.4 Gy	60	surgical tumor material, pre-treatment tumor material	1	*KRAS*	TaqMan	pCR	pCR (pCR or, in patients who did not undergo surgery, radiologic CR) 14%	*KRAS* - rs61764370 TG
Sclafani [39]	2016	Retrospective cohort	UK, Spain, Sweden, and others	155	2005–2008	staging not specified	CAPOX (±Cetuximab) + Capecitabine-RT 45 Gy + 5.4 Gy	60	surgical tumor material, pre-treatment tumor material	1	*miR-608*	TaqMan	pCR	pCR (pCR or, in patients who did not undergo surgery, radiologic CR) 14%	none
Sebio [40]	2015	Retrospective cohort	Spain	84	NR	stage II (27.4%), stage III (72.6%)	RT 45 Gy (1.8 Gy per fraction) + Capecitabine	68	peripheral blood sample	28	*TYMS*, *XRCC1*, *ERCC1*, *AREG*, *EGF*, *EREG*, *EGFR*, *ERCC2* (*XPD*)	TaqMan	pCR	complete response (Mandard TRG 1) 20.2%	*ERCC1* - rs11615 CT/TT *AREG* - rs11942466 CC (wild type)
Spindler [41]	2006	Prospective cohort	Denmark	77	2003–2005	T3N0M0 (22%) T3N1M0 (62%) T3N2M0 (16%)	RT 65 Gy + UFT + Leucovorin	64	peripheral blood sample	1	*EGFR*	TaqMan	Mandard TRG	responders (TRG 1–2) 49% non-responders (TRG 3–4) 51%	*EGFR* - rs712829 GT/TT
Stanojevic [42]	2024	Prospective cohort	Serbia	97	2018–2019	stage II (8%), stage III (92%)	RT 50.4 Gy (1.8 Gy per fraction) + 5-FU + Leucovorin	61	pre-treatment tumor material	2	*MTHFR*	PCR-RFLP	Mandard TRG	responders (TRG 1–2) 31% non-responders (TRG 3–5) 67% NR 2%	none
Stoehlmacher [43]	2008	Retrospective cohort	Germany	40	1998–2001	stage II or III (100%)	RT 50.4 Gy (1.8 Gy per fraction) + 5-FU	60	pre-treatment tumor material	1	*TYMS*	PCR-RFLP	TRG *	responders 84% non-responders 16%	none
Terrazzino [44]	2006	Retrospective cohort	Italy	125	1994–2002	T2 (8%), T3 (66%), T4 (24%); N1 (67%); M0 (100%)	RT 48.4 Gy (median) + 5-FU (35%)/5-FU + Oxaliplatin (22%)/Leucovorin (29%)/Carboplatin (14%)	60	peripheral blood sample	2	*MTHFR*	PCR-RFLP	Mandard TRG	responders (TRG 1–2) 39% non-responders (TRG 3–4) 61%	*MTHFR* - rs1801133 CC
Xiao [45]	2016	Prospective cohort	China	58	2007–2012	stage II (22%), stage III (78%)	RT 46 Gy (2 Gy per fraction) + XELOX (86%) or mFOLFOX6 (10%)	NR	pre-treatment tumor material	1	*IL13*	Sanger sequencing	pCR, Dworak TRG	pCR (Dworak TRG 4) 28% good response (TRG 3–4) 48% non-responders (TRG 0–2) 52%	none

* grade 0 = no regression; grade 1 = dominant tumor mass with obvious fibrosis or mucin; grade 2 = dominantly fibrotic or mucinous changes, with few tumor cells or groups; grade 3 = very few tumor cells in fibrotic or mucinous tissue; grade 4 = no tumor cells. NR—not reported.

**Table 3 cancers-17-03995-t003:** Genes and corresponding SNPs analyzed in the included studies.

Gene Function	Name	Studies	Number of Studies	SNP ID	Number of Different SNPs Analyzed
Folate metabolism	*DPYD*	Balboa [15]	1	rs3918290	1
	*MTHFR*	Balboa [15], Boige [16], Cecchin [17], Dreussi [19], Formica [22], Garcia-Aguilar [23], Ho-Pun-Cheung [26], Hu-Lieskovan [28], Lamas [31], Nikas [34], Stanojevic [42], Terrazzino [44]	12	rs3737967, rs3818762, rs3737964, rs7553194, rs17367504, rs9651118, rs4846052, rs1572151, rs1801133, rs1801131, rs17375901	11
	*UMPS*	Kim [30]	1	rs1801019	1
	*TYMS*	Balboa [15], Boige [16], Hur [29], Lamas [31], Páez [35], Sebio [40], Stoehlmacher [43]	7	rs2853542, rs2847153, rs2298582, rs2612101, rs10502290, rs2260821, rs3744962, rs1001761, rs2853741, VNTR/5′UTR	11
DNA repair	*ERCC5*	Ho-Pun-Cheung [26]	1	rs17655	1
	*MSH6*	Dreussi [19]	1	rs3136228	1
	*CHEK2*	Ho-Pun-Cheung [26]	1	rs2267130	1
	*RAD51*	Cecchin [17], Hu-Lieskovan [28]	2	rs1801320, rs5030783, rs1801321	3
	*ERCC1*	Balboa [15], Boige [16], Cecchin [17], Dreussi [19], Formica [22], Ho-Pun-Cheung [26], Lamas [31], Páez [35], Sebio [40]	9	rs11615, rs10412761, rs2336219, rs3212986, rs2298881, rs4803823, rs3212948	7
	*MGMT*	Dreussi [19], Ho-Pun-Cheung [26]	2	rs12917	1
	*EXO1*	Dreussi [19]	1	rs4149963	1
	*MLH1*	Cecchin [17], Dreussi [19]	2	rs1799977, rs1800734	2
	*MSH2*	Cecchin [17]	1	rs2303428	1
	*XRCC1*	Balboa [15], Boige [16], Cecchin [17], Dreussi [19], Formica [22], Grimminger [24], Ho-Pun-Cheung [26], Lamas [31], Nicosia [33], Páez [35], Sebio [40]	11	rs25487, rs2293036, rs3213334, rs2023614, rs1001581, rs2854496, rs3213266, rs3213255, rs304729, rs1799782, rs3213239, rs25489, rs861539, rs3213245	14
	*BAX*	Ho-Pun-Cheung [26]	1	rs36017265, rs4645878	2
	*XPA*	Boige [16], Ho-Pun-Cheung [26]	2	rs2773354, rs3176757, rs2808667, rs2805835, rs3176689, rs3176683, rs3176658, rs3176639, rs1800975	9
	*XRCC3*	Boige [16], Cecchin [17], Dreussi [19], Ho-Pun-Cheung [26], Hu-Lieskovan [28]	5	rs3212102, rs12432907, rs3212090, rs3212079, rs861531, rs861530, rs861528, rs1799794, rs861539, rs1799796	10
	*PARP-1*	Dreussi [19], Ho-Pun-Cheung [26]	2	rs11136410	1
	*ADPRT*	Ho-Pun-Cheung [26]	1	rs1136410	1
	*ERCC4*	Boige [16], Ho-Pun-Cheung [26]	2	rs1364362, rs1800067, rs11075223, rs1799802, rs744154, rs1799801, rs1799800	7
	*LIG4*	Ho-Pun-Cheung [26]	1	rs1805388, rs1805386	2
	*CHEK1*	Ho-Pun-Cheung [26]	1	rs521102	1
	*LIG3*	Ho-Pun-Cheung [26]	1	rs1052536, rs3135967	2
	*ERCC2* (*XPD*)	Balboa [15], Boige [16], Cecchin [17], Dreussi [19], Ho-Pun-Cheung [26], Páez [35], Sebio [40]	7	rs13181, rs238415, rs50872, rs50871, rs1799793, rs11878644, rs28365048, rs1799787	9
	*MBD4*	Ho-Pun-Cheung [26]	1	rs10342, rs140693	2
	*APEX1*	Dreussi [19], Ho-Pun-Cheung [26]	2	rs1130409, rs1760944	2
	*NBN*	Ho-Pun-Cheung [26]	1	rs1805794	1
	*RECQL*	Ho-Pun-Cheung [26]	1	rs13035	1
	*ZNF350*	Ho-Pun-Cheung [26]	1	rs2278415, rs2278420	2
	*RAD52*	Ho-Pun-Cheung [26]	1	rs11226	1
	*XRCC5*	Ho-Pun-Cheung [26]	1	rs1051677, rs1051685, rs6941, rs2440	4
	*PTEN*	Peng [36]	1	rs2299939, rs12569998	2
Immune regulation	*TNFA*	Ho-Pun-Cheung [26]	1	rs1800629	1
	*LTA*	Ho-Pun-Cheung [26]	1	rs2229094	1
	*IL8*	Ho-Pun-Cheung [26], Hu-Lieskovan [28]	2	rs4073	1
	*IL4*	Ho-Pun-Cheung [26]	1	rs2243250	1
	*FPR1*	Chiang [18]	1	rs867228	1
	*IL13*	Ho-Pun-Cheung [26], Xiao [45]	2	rs20541, rs1800925	2
	*IL10*	Ho-Pun-Cheung [26]	1	rs1800896	1
	*FAM101A*	Kim [14]	1	rs7955740	1
	*TLR1*	Chiang [18]	1	rs5743618	1
	*IL6*	Ho-Pun-Cheung [26]	1	rs1800795	1
	*P2RX7*	Chiang [18]	1	rs3751143	1
	*NFKB1*	Ho-Pun-Cheung [26]	1	rs3774932, rs3774934, rs3774936, rs3774937	4
	*FCGR2A*	Ho-Pun-Cheung [26], Hu-Lieskovan [28]	2	rs1801274	1
	*FCGR3A*	Ho-Pun-Cheung [26], Hu-Lieskovan [28]	2	rs396991	1
	*IL1B*	Dzhugashvili [21], Ho-Pun-Cheung [26]	2	rs16944, rs1143627, rs1143634	3
	*TIM3*	Chiang [18]	1	rs1036199	1
	*CORO2A*	Kim [14]	1	rs1985859	1
Angiogenesis	*HIF-1A*	Havelund [25], Ho-Pun-Cheung [26]	2	rs11549465, rs11549467, rs2057482, rs2246350	4
	*VEGF*	Dreussi [19], Ho-Pun-Cheung [26], Hu-Lieskovan [28]	3	rs2010963, rs1570360, rs3025039, rs699947	4
Growth factor receptor	*IGF2R*	Ho-Pun-Cheung [26]	1	rs629849	1
	*ERBB2*	Ho-Pun-Cheung [26]	1	rs1801200	1
	*EGFR*	Balboa [15], Dreussi [19], Ho-Pun-Cheung [26], Hu-Lieskovan [28], Páez [35], Sebio [40], Spindler [41]	7	rs11568315, rs2227983, rs17290169, rs17335738, rs712830, rs712829, rs11543848	7
	*EGF*	Dreussi [19], Ho-Pun-Cheung [26], Hu-Lieskovan [28], Sebio [40]	4	rs4444903, rs6533485, rs11568993, rs4698803, rs11568972, rs929446, rs2074390, rs6850557	8
	*EREG*	Sebio [40]	1	rs7687621, rs1017733	2
	*TGFB1*	Ho-Pun-Cheung [26]	1	rs1982073, rs1800471, rs1800469	3
	*IGF1*	Ho-Pun-Cheung [26]	1	rs2229765	1
	*FGFR4*	Ho-Pun-Cheung [26]	1	rs351855	1
	*FGF2*	Ho-Pun-Cheung [26]	1	rs308447	1
	*AREG*	Sebio [40]	1	rs11942466, rs28635876, rs13104811, rs1353295, rs3913032, rs6447003, rs10034692, rs11725706, rs2132065	9
Oncogene	*BRCA2*	Ho-Pun-Cheung [26]	1	rs1799943, rs206143, rs144848	3
	*BRCA1*	Ho-Pun-Cheung [26]	1	rs1799966, rs16941, rs16942, rs799917	4
	*KRAS*	Hu-Lieskovan [28], Sclafani [38]	2	rs61764370	1
	*PIK3CA*	Peng [36]	1	rs2699887, rs6443624, rs7621329, rs7651265	4
miRNA	*miR-371a*	Dreussi [20]	1	rs28461391	1
	*SMAD5*	Dreussi [20]	1	rs1057898, rs6871224	2
	*DDX20*	Dreussi [20]	1	rs197412	1
	*TNRC6B*	Dreussi [20]	1	rs139911	1
	*miR-608*	Sclafani [39]	1	rs4919510	1
	*DGCR8*	Dreussi [20]	1	rs417309	1
	*CNOT4*	Dreussi [20]	1	rs11772832	1
	*TNRC6A*	Dreussi [20]	1	rs6497759	1
	*TRBP*	Dreussi [20]	1	rs6088619	1
	*miR-196a-2*	Dreussi [20]	1	rs11614913	1
	*SMAD2*	Dreussi [20]	1	rs1792671	1
	*CNOT6*	Dreussi [20]	1	rs6877400	1
	*DICER1*	Dreussi [20]	1	rs1057035	1
	*DROSHA*	Dreussi [20]	1	rs10719	1
Transporter	*SLC10A7*	Kim [14]	1	rs41398848	1
Tumor suppressor	*TP53*	Dreussi [19], Ho-Pun-Cheung [26]	2	rs1642785, rs1042522, rs2602141, rs560191	4
Drug transporters	*ABCB1*	Cecchin [17], Formica [22], Kim [30]	3	rs1045642, rs1128503, rs2032582	3
	*CYP2A6*	Kim [30]	1	rs5031016, rs28399433, rs28399468	4
	*ABCC2*	Cecchin [17]	1	rs2273697, rs717620	2
Detoxication	*GSTA1*B*	Cecchin [17]	1	rs3957357	1
	*GSTP1*	Boige [16], Cecchin [17], Dreussi [19], Formica [22], Ho-Pun-Cheung [26], Nicosia [33], Páez [35]	7	rs7927381, rs6591256, rs1138272, rs947894, rs1695	6
	*CYP1A1*	Ho-Pun-Cheung [26]	1	rs1048943	1
	*GSTT1*	Ho-Pun-Cheung [26]	1	rs4630	1
Reactive oxygen species	*SOD2*	Boige [16], Dreussi [19], Ho-Pun-Cheung [26], Leu [32]	4	rs5746136, rs5746141, rs2842980, rs2758329, rs4342445, rs4880	7
	*MPO*	Ho-Pun-Cheung [26], Leu [32]	2	rs7208693, rs2333227	2
	*SOD3*	Leu [32]	1	rs699473	1
	*NOS2A*	Ho-Pun-Cheung [26]	1	rs2297518	1
	*CAT*	Leu [32]	1	rs1001179, rs769214	2
	*MT-ND3*	Ho-Pun-Cheung [26]	1	rs2853826	1
	*NOS3*	Ho-Pun-Cheung [26]	1	rs179998	1
	*GPX1*	Ho-Pun-Cheung [26], Leu [32]	2	rs1050450	1
	*OGG1*	Cecchin [17], Dreussi [19], Leu [32]	3	rs1052133	1
	*CYBA*	Leu [32]	1	rs1049255	1
Cell cycle regulator	*AKT1*	Peng [36]	1	rs1130214, rs2494738, rs2498804	3
	*TP73*	Ho-Pun-Cheung [26]	1	rs2273953, rs1801173	2
	*CDKN1A*	Ho-Pun-Cheung [26]	1	rs1801270	1
	*CCND1*	Garcia-Aguilar [23], Ho-Pun-Cheung [26], Ho-Pun-Cheung [27], Hu-Lieskovan [28]	4	rs603965, rs9344	2
	*ATM*	Dreussi [19], Ho-Pun-Cheung [26]	2	rs1801516, rs189037, rs1800057	3
	*MDM2*	Dreussi [19], Ho-Pun-Cheung [26]	2	rs2279744, rs1470383	2
	*AKT2*	Peng [36]	1	rs8100018	1
	*FRAP1*	Peng [36]	1	rs2295080, rs11121704	2
	*TERT*	Rampazzo [37]	1	rs2736108, rs2735940, rs2736098, rs2736100, rs35241335, rs11742908, rs2736122, rs2853690	8
Other	*CDC42BPA*	Kim [14]	1	rs192986	1
	*ASZ1*	Kim [14]	1	rs7808424	1
	*SMAD3*	Dreussi [20]	1	rs17228212, rs2289791, rs744910, rs745103, rs8025774, rs8028147	6
	*OR2T4*	Kim [14]	1	rs1538704	1
	*USP20*	Kim [14]	1	rs2274507	1
	*ICAM5*	Ho-Pun-Cheung [26]	1	rs1056538, rs2228615	2
Apoptosis	*BCL2*	Ho-Pun-Cheung [26]	1	rs2279115	1
	*CASP9*	Ho-Pun-Cheung [26]	1	rs1052576	1
	*CASP8*	Ho-Pun-Cheung [26]	1	rs1045485, rs13113	2
	*CASP10*	Ho-Pun-Cheung [26]	1	rs13010627	1
	*CASP3*	Ho-Pun-Cheung [26]	1	rs6948, rs1049216	2
Transcription factors	*ZNF281*	Kim [14]	1	rs4244146	1
	*PPARG*	Ho-Pun-Cheung [26]	1	rs1801282	1
	*MED4*	Kim [14]	1	rs1571256	1
	*NFE2L2*	Ho-Pun-Cheung [26]	1	rs5031039, rs35652124	3
β-catenin pathway	*GSK3B*	Ho-Pun-Cheung [26]	1	rs334558, rs3755557, rs6721961	2
	*CTNNB1*	Ho-Pun-Cheung [26]	1	rs4135385, rs13072632	2
Cyclooxygenase	*PTGS2*	Dzhugashvili [21], Ho-Pun-Cheung [26], Hu-Lieskovan [28]	3	rs5275, rs20417	2
	*PTGS1*	Dzhugashvili [21]	1	rs1213266, rs5789	2

**Table 4 cancers-17-03995-t004:** Summary of SNPs related to treatment response in at least one study.

Name	Genes	Number of Studies with Effect/All Studies	Studies Showing Significance	Studies Not Showing Significance	Number of Patients in Studies Showing Significance/Total Number of Patients	Positive Predictors of Pathological Response	Allele Frequency	Effect Size *
Folate metabolism pathways								
rs2853542 G>C	*TYMS*	1/6	Lamas [31]	Balboa [15], Hur [29], Páez [35], Sebio [40], Stoehlmacher [43]	93/454	rs2853542 2R/3G, 3C/3G, 3G/3G	39%	OR: 2.65; 95% CI: 1.10–6.39, *p* = 0.02
rs1801133 C>T	*MTHFR*	3/10	(1) Cecchin [17], (2) Nikas [34], (3) Terrazzino [44]	Boige [16], Dreussi [19], Garcia-Aguilar [23], Ho-Pun-Cheung [26], Hu-Lieskovan [28], Lamas [31], Stanojevic [42]	471/1590	(1,2,3) rs1801133 CC	(1) 64% (2) 54% (3) 33%	(1) OR: 2.00; 95% CI: 1.03–4.00; *p* < 0.05 (2) OR: 2.91; 95% CI: 1.23–6.89; *p* = 0.0150 (3) OR: 3.13; 95% CI: 1.39–7.14; *p* = 0.002
DNA repair pathway								
rs25487 A>G	*XRCC1*	4/10	(1) Balboa [15], (2) Formica [22], (3) Grimminger [24], (4) Lamas [31]	Cecchin [17], Dreussi [19], Ho-Pun-Cheung [26], Nicosia [33], Páez [35], Sebio [40]	239/1120	(1) rs25487 AA, (2) rs25487 GG, (3) rs25487 AG, (4) rs25487 GG	(1) 10% (2) NR (3) 40% (4) 47%	(1) OR: 7.93 95% CI: 1.03–60.83; *p* = 0.006 (2) OR: 25.8; 95% CI: 1.02–653.85; *p* = 0.049 (3) OR: NR, 95% CI: NR; *p* = 0.039 (4) GG vs. GA: OR: 4.18; 95% CI: 1.62–10.74; *p* = 0.003
rs11615 T>C	*ERCC1*	1/8	Sebio [40]	Balboa [15], Cecchin [17], Dreussi [19], Formica [22], Ho-Pun-Cheung [26], Lamas [31], Páez [35]	84/959	rs11615 TT	20%	TT vs. CT OR: 2.27; 95% CI: 0.76–7.69; *p* = 0.0235
rs10412761 A>G	*ERCC1*	1/1	Boige [16]		316/361	rs10412761 AG/GG	44%	OR: 1.75, 95% CI: 1.02–2.94, *p* = 0.042
rs1799787 C>T	*ERCC2* (*XPD*)	1/1	Boige [16]		316/361	rs1799787 CT/TT	44%	OR: 1.82, 95% CI: 1.08–3.13, *p* = 0.027
Reactive oxygen species pathways								
rs1695 A>G	*GSTP1*	1/5	Nicosia [33]	Dreussi [19], Formica [22], Ho-Pun-Cheung [26], Páez [35]	80/559	rs1695 AA	NR	OR: NR, 95% CI: NR; *p* = 0.04
rs4880 C>T	*SOD2*	1/3	Ho-Pun-Cheung [26]	Dreussi [19], Leu [32]	71/638	rs4880 CC	32%	OR: 5.26; 95% CI: 1.56–16.67; *p* = 0.005
rs1052133 C>G	*OGG1*	1/4	Cecchin [17]	Dreussi [19], Ho-Pun-Cheung [26], Leu [32]	238/876	rs1052133 CC	69%	OR: 2.13; 95% CI: 1.06–4.17; *p* < 0.05
Growth factor receptor pathways								
rs4444903 A>G	*EGF*	1/4	Hu-Lieskovan [28]	Dreussi [19], Ho-Pun-Cheung [26], Sebio [40]	130/565	rs4444903 AG/GG	54%	OR: 16.68; 95% CI: 2.1–130.8; *p* = 0.007
rs712829	*EGFR*	1/3	Spindler [41]	Ho-Pun-Cheung [26], Sebio [40]	77/232	rs712829 GT/TT	54%	OR: NR, 95% CI: NR; *p* = 0.023
rs11942466 C>A	*AREG*	1/1	Sebio [40]		84/84	rs11942466 CC	20%	CC vs. CA OR: 2.33; 95% CI: 0.75–7.14; *p* = 0.0018
Cell cycle regulator pathways								
rs603965 G>A	*CCND1*	1/3	Ho-Pun-Cheung [27]	Garcia-Aguilar [23], Ho-Pun-Cheung [26]	70/273	rs603965 AA	14%	OR: 10.0; 95% CI: 1.2–84.7; *p* = 0.034
Immune regulation pathways								
rs1985859 C>T	*CORO2A*	1/1	Kim [14]		113/113	rs1985859 CC	34%	OR: 4.88; 95% CI: 1.06–22.73; *p* = 0.03
rs867228 T>C	*FPR1*	1/1	Chiang [18]		130/130	rs867228 AC/AA	42%	OR: 2.521; 95% CI: 1.162–5.473; *p* = 0.017
rs1800925 C>T	*IL13*	1/2	Ho-Pun-Cheung [26]	Xiao [45]	71/129	rs1800925 CC	63%	OR: 7.14; 95% CI: 2.04–25.00; *p* = 0.0008
Oncogenic pathways								
rs61764370 T>G	*KRAS*	1/2	Sclafani [38]	Hu-Lieskovan [28]	155/285	rs61764370 TG	21%	OR: NR, 95% CI: NR; *p* = 0.02
Telomere length pathways								
rs2736108 C>T	*TERT*	1/1	Rampazzo [37]		194/194	rs2736108 CC	NR	CC vs. TT OR: 4.6; 95% CI: 1.1–19.1; *p* = 0.034
rs2853690 G>A	*TERT*	1/1	Rampazzo [37]		194/194	rs2853690 GG/AA	NR	AA/GG vs. AG OR: 3.0; 95% CI: 1.3–6.9; *p* = 0.008
Other pathways								
rs17228212 C>T	*SMAD3*	1/1	Dreussi [20]		265/265	rs17228212 TT	NR	OR: 2.01; 95% CI: 1.22–3.31; *p* = 0.0064
rs744910 A>G	*SMAD3*	1/1	Dreussi [20]		265/265	rs744910 AG/GG	NR	OR: 2.22; 95% CI: 1.18–4.17; *p* = 0.0135
rs745103 T>C	*SMAD3*	1/1	Dreussi [20]		265/265	rs745103 GG	NR	OR 2.08; 95% CI: 1.06–4.00; *p* = 0.0316
rs6088619 C>T	*TRBP*	1/1	Dreussi [20]		265/265	rs6088619 AG/GG	NR	OR: 2.56; 95% CI: 1.27–5.26; *p* = 0.0089
rs10719 T>C	*DROSHA*	1/1	Dreussi [20]		265/265	rs10719 CC	NR	OR: 3.0; 95% CI: 1.3–6.9; *p* = 0.008

* Effect size refers to comparisons between the indicated genotype(s)/variant(s) and all other genotypes combined, unless specified otherwise. OR—odds ratio; CI—confidence interval; NR—not reported.

**Table 5 cancers-17-03995-t005:** Risk of bias in the included studies according to the QGenie tool.

Author	Year	Rationale for Study	Selection and Definition of Outcome of Interest	Selection and Comparability of Comparison Groups (If Applicable)	Technical Classification of the Exposure	Non-Technical Classification of the Exposure	Other Sources of Bias	Sample Size and Power	A Priori Planning of Analyses	Statistical Methods and Control for Confounding	Testing of Assumptions and Inferences for Genetic Analyses	Appropriateness of Inferences Drawn from Results	Overall
Balboa [15]	2010	6	6	4	6	5	5	3	5	6	5	6	57/77
Boige [16]	2019	6	6	5	6	6	5	4	6	6	4	4	58/77
Cecchin [17]	2011	6	6	4	6	5	5	5	5	5	5	6	58/77
Chiang [18]	2021	6	6	5	6	4	6	4	5	5	3	6	56/77
Dreussi [20]	2016	6	6	5	6	5	4	5	5	5	5	6	58/77
Dreussi [19]	2016	6	6	5	5	4	4	5	5	6	3	6	55/77
Dzhugashvili [21]	2014	6	6	4	6	4	5	5	5	6	4	6	57/77
Formica [22]	2018	5	6	4	3	2	4	2	5	6	3	6	46/77
Garcia-Aguilar [23]	2011	6	6	5	6	5	5	5	5	6	3	6	58/77
Grimminger [24]	2010	6	6	4	6	5	4	3	5	6	3	6	54/77
Havelund [25]	2012	6	6	6	6	5	3	4	5	5	4	6	56/77
Ho-Pun-Cheung [27]	2011	6	6	5	5	4	3	3	5	6	3	6	52/77
Ho-Pun-Cheung [26]	2007	5	6	4	5	5	3	2	5	6	3	5	49/77
Hu-Lieskovan [28]	2011	6	6	3	4	5	4	4	5	6	3	5	51/77
Hur [29]	2011	6	6	5	4	6	2	2	5	6	3	4	49/77
Kim [14]	2013	6	6	4	5	3	3	4	4	6	4	6	51/77
Kim [30]	2017	5	5	5	3	3	6	4	5	5	5	5	51/77
Lamas [31]	2012	6	6	5	4	3	3	4	5	6	4	6	52/77
Leu [32]	2021	7	6	5	4	3	5	6	5	7	4	7	59/77
Nicosia [33]	2018	5	6	5	6	3	2	3	5	5	4	5	49/77
Nikas [34]	2015	7	7	4	4	5	1	5	6	5	2	6	52/77
Páez [35]	2011	5	6	4	5	5	3	3	4	4	4	5	48/77
Peng [36]	2018	6	5	4	4	3	4	3	5	5	3	6	48/77
Rampazzo [37]	2020	6	4	3	5	3	2	3	5	4	4	4	43/77
Sclafani [38]	2015	5	3	4	5	5	3	3	4	4	5	6	47/77
Sclafani [39]	2016	7	6	6	6	7	4	3	6	5	6	6	62/77
Sebio [40]	2015	6	4	4	6	2	2	3	6	5	5	4	47/77
Spindler [41]	2006	6	3	4	3	2	3	2	2	3	2	4	34/77
Stanojevic [42]	2024	6	7	7	7	5	5	3	6	5	7	7	65/77
Stoehlmacher [43]	2008	6	3	3	4	2	4	1	4	3	2	6	38/77
Terrazzino [44]	2006	6	6	4	6	5	2	3	6	5	6	5	54/77
Xiao [45]	2016	5	3	2	3	2	5	2	6	3	4	6	41/77
Criterium average		5.88	5.50	4.41	5.00	4.09	3.72	3.47	5.00	5.19	3.91	5.56	

Green means the highest score (lowest risk of bias) and red means the lowest score (highest risk of bias).

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
