# Peer review of "Single Nucleotide Polymorphisms as Biomarkers of Response to Neoadjuvant Chemoradiotherapy in Rectal Cancer: A Systematic Review"

_cancers, 2025, doi:10.3390/cancers17243995_

Round 1

Reviewer 1 Report

Comments and Suggestions for Authors

The manuscript presents a comprehensive and well-structured synthesis of a field that is often fragmented and inconclusive, yet carries significant clinical implications. The investigation of single nucleotide polymorphisms (SNPs) as predictive biomarkers of response to neoadjuvant chemoradiotherapy (nCRT) in rectal cancer is not a new topic; this area has been studied for over a decade, with several narrative and smaller systematic reviews published in recent years. Nevertheless, the authors distinguish themselves by offering a thorough and organized synthesis of the scattered evidence.

This review incorporates data from 32 studies encompassing several thousand patients, representing a substantial expansion in scope compared with earlier reviews, which typically included fewer studies. The sheer number of SNPs assessed in a single review is exceptional, providing a more comprehensive overview than previous work. Employing a structured, PRISMA-based methodology with explicit risk-of-bias assessment enhances the rigor relative to some earlier narrative reviews. While the conclusion that no SNP demonstrates robust predictive value is neither new nor surprising, this study strengthens that assertion by supporting it with a large and diverse dataset.

The Discussion section could be further refined. It appropriately situates the findings within the broader literature, citing previous systematic reviews and meta-analyses, but it mainly emphasizes the lack of association for any SNP without discussing potential positive versus negative associations, or trends that may have emerged. Additionally, the Conclusion section is currently missing.

Overall, although the study does not present novel biological discoveries, it stands out as one of the most comprehensive and methodologically rigorous systematic evaluations of SNP predictors in rectal cancer nCRT to date. Its novelty lies in its breadth, clarity of synthesis, and updated evidence summary, providing a valuable reference and helping to guide future research in this field.

Author Response

Comments 1: The Discussion section could be further refined. It appropriately situates the findings within the broader literature, citing previous systematic reviews and meta-analyses, but it mainly emphasizes the lack of association for any SNP without discussing potential positive versus negative associations, or trends that may have emerged. 

Response 1: Thank you you for pointing this out. We agree with this comment. Therefore, we have discussed single positive associations in SNPs in genes GSTP1, EGF, IL13, FPR1 and TERT (page 9, line 186-190), both negative and positive associations in XRCC1 rs25487 and MTHFR rs1801133 (page 9, line 194-199).

Comments 2: Additionally, the Conclusion section is currently missing.

Response 2: Thank you for pointing this out. Therefore, we have added a Conclusion section (page 11, line 283-301).

Reviewer 2 Report

Comments and Suggestions for Authors

The manuscript titled “Single nucleotide polymorphisms as biomarkers of response to neoadjuvant chemoradiotherapy in rectal cancer: a systematic review” is well written and presented.

This study shows that the current evidence does not support using individual SNPs to reliably predict nCRT response in rectal cancer. Although XRCC1 and MTHFR are among the most studied polymorphisms, their predictive value remains inconsistent and inconclusive.

This manuscript is recommended for acceptance after a minor revision to improve clarity and incorporate specific feedback, authors please provide explanations:

  1. There is a significant heterogeneity in treatment protocols, radiation doses, response assessment methods, and definitions of pathological response across studies.
  2. The current study has small sample sizes with limited statistical power, increasing the risk of false-positive or non-replicable results.
  3. Most reported associations came from single studies without independent validation cohorts.
  4. There is variability in genotyping platforms and lack of standardized quality-control reporting may introduce technical errors.
  5. In this study, most analyses focused on single SNPs rather than polygenic models or gene–gene and gene–environment interactions.
  6. There is limited ethnic and demographic diversity which restrict generalizability of the findings.

Author Response

Comments 1: There is a significant heterogeneity in treatment protocols, radiation doses, response assessment methods, and definitions of pathological response across studies.

Response 1: Thank you for pointing this out. As this is a systematic review of 32 independent genetic association studies, differences in neoadjuvant regimens (dose and fractionation of radiotherapy, type and combinations of fluoropyrimidines), TNT regimens, and TRG scales (Mandard, Dworak and AJCC), as well as definitions of pathological response, are largely unavoidable. This heterogeneity is emphasized more clearly in the updated manuscript's “Results” section (page 5, line 151-161; page 6, line 181-190) and “Discussion” section (page 9, line 203-204), which note that it limits the feasibility of quantitative meta-analyses and comparisons of effects between studies.

Comments 2: The current study has small sample sizes with limited statistical power, increasing the risk of false-positive or non-replicable results.

Response 2:Thank you for pointing this out. We agree that most primary studies were underpowered, as the risk of bias analysis also highlighted (average score of 3.47/7 in the 'sample size and power' domain). The 'Discussion' section (page 10, line 232-236) now states that small sample sizes combined with multiple testing increase the risk of false-positive results, which may explain why the reported associations are not replicable.

Comments 3: Most reported associations came from single studies without independent validation cohorts.

Response 3: We agree that most of the reported associations originate from single studies that have not been independently validated. We have  commented on the fact that most associations come from single studies (page 9, lines 186-189) and added a paragraph (page 9, line 194-199) emphasizing that, even for the most widely studied SNPs (XRCC1 rs25487 and MTHFR rs1801133), the number of patients in negative studies significantly outweighs the number in positive studies. This currently precludes clinical use.

Comments 4: There is variability in genotyping platforms and lack of standardized quality-control reporting may introduce technical errors.

Response 4: The methodological remarks were expanded to describe the differences between the genotyping techniques used (PCR-RFLP, TaqMan, Sanger sequencing, SNP array, SNaPshot and MassARRAY), and to emphasize that detailed quality control data (e.g. reproducibility, call rate and positive/negative controls) were often sparsely or not reported (page 9, line 105-218). The lack of standardization of these elements was emphasized as it may introduce technical errors and further hinder cross-study comparisons.

Comments 5: In this study, most analyses focused on single SNPs rather than polygenic models or gene–gene and gene–environment interactions.

Response 5: Most of the included studies applied classical candidate SNP approaches, analyzing single variants in isolation and rarely considering gene–gene or gene–environment interactions. The 'Discussion' section now clearly states that future research should employ polygenic models and integrate multi-omics data layers (e.g. genomics, transcriptomics and proteomics) to improve predictive power. (page 10-11, line 266-276)

Comments 6: There is limited ethnic and demographic diversity which restrict generalizability of the findings.

Response 6: As shown in Table 2, the majority of studies included European and some studies East Asian populations; with other ethnic groups underrepresented. The Discussion (page 10, line 218-222) clarifies that this lack of diversity, together with potential population-specific effects such as differences in allele frequency and linkage disequilibrium (LD) structure, limits generalisability and may partially explain replication failures across populations.

Reviewer 3 Report

Comments and Suggestions for Authors

The manuscript by Polomskaya et al. addresses a clinically related issue and provides a detailed review of current data on predictors of response to neoadjuvant chemoradiotherapy (nCRT) in rectal cancer based on single nucleotide polymorphisms (SNPs). This topic is timely and of interest to clinicians and researchers seeking accurate tools for treatment stratification.

However, there are key areas in which the manuscript requires clarification, addition, or expansion before it can be considered for publication.

Detailed comments

Major Comments

  1. Definition of “Response” requires clearer normalization

Across studies, pCR, TRG, and downstaging are used interchangeably. The manuscript should present:

A table harmonizing response definitions, the frequency of each endpoint in the included studies, and a clearer discussion of how endpoint variability impacts synthesizability.

  1. Biological and clinical integration needs strengthening

Although many polymorphisms are listed, the manuscript would gain value from deeper mechanistic interpretation:

  • Why are XRCC1, XRCC3, RAD51, and MGMT repeatedly implicated?

How do germline variants compare with somatic tumor markers (e.g., KRAS, MSI status)?

       2)    Where do these germline markers fit into current predictive frameworks such as radiomics or gene-expression signatures?

  1. Risk of bias and study quality assessment underdeveloped

The manuscript should include:

  • a structured risk-of-bias table (e.g., using QUADAS-2 or equivalent),
  • assessment of how study quality influences conclusions,
  • and whether low-quality studies tend to report more positive findings.
  1. Clinical translation section needs expansion

The conclusion states that no SNP is ready for clinical use, which is appropriate; however:

  • A more explicit translational roadmap is warranted.
  • Consider outlining requirements for future biomarker trials (sample size, standardized endpoints, multi-SNP panels, external validation).
  1. Tables summarizing SNPs could be improved

Some tables are difficult to interpret quickly. Consider:

o     grouping SNPs by pathway,

o     including allele frequencies,

o     and adding effect directionality where available.

Minor comments

  1. Typographical and stylistic corrections

A small number of sentences could be rephrased for greater precision and brevity.

  1. MGMT

Since this gene is involved in epigenetic process, its inclusion should be conceptually separated.

The conclusion must be included in the manuscript.

Author Response

Comments 1: Across studies, pCR, TRG, and downstaging are used interchangeably. The manuscript should present a table harmonizing response definitions, the frequency of each endpoint in the included studies, and a clearer discussion of how endpoint variability impacts synthesizability.

Response 1: Thank you for highlighting this important issue. We fully agree with the reviewer’s comment. In the revised manuscript, we have harmonized the outcome definitions and incorporated them into Table 2, specifically in the column Responder/pCR definition and rate, which now provides unified definitions and the frequency of each endpoint across studies. These data are summarised in the Results section (page 6, lines 181–190), and the implications of outcome heterogeneity for evidence synthesis are discussed in the Discussion section (page 10, lines 203–204).

Comments 2: Although many polymorphisms are listed, the manuscript would benefit from deeper mechanistic interpretation:

(a) Why are XRCC1, XRCC3, RAD51, and MGMT repeatedly implicated?

(b) How do germline variants compare with somatic tumor markers (e.g., KRAS, MSI status)? (

c) Where do these germline markers fit within current predictive frameworks such as radiomics or gene-expression signatures?

Response 2: Thank you for these insightful remarks.

(a) We agree that the recurrence of certain polymorphisms warrants clarification. XRCC1, XRCC3, MTHFR, ERCC1, ERCC2, TYMS, and EGFR appear repeatedly because most included studies used a candidate-gene approach focusing on biological pathways relevant to radiosensitivity and fluoropyrimidine metabolism. We have expanded this explanation in the Discussion section (page 9, lines 179–190). We also note that RAD51 and MGMT were examined in only two studies each.

(b) We now compare germline and somatic biomarkers in the Discussion section (page 10, lines 256–262), summarizing evidence on KRAS and MSI as clinically validated predictors of response to nCRT, and emphasising the contrast with the limited and inconsistent data on germline SNPs.

(c) We have added a paragraph outlining how future predictive frameworks should integrate germline variants with radiomics, multi-omics signatures, and machine-learning approaches (page 10–11, lines 267–277).

Comments 3: The manuscript should include a structured risk-of-bias table, an assessment of how study quality influences conclusions, and a discussion on whether low-quality studies tend to report more positive findings.

Response 3: We appreciate this recommendation. The risk-of-bias evaluation (QGenie tool) has now been moved from the Supplement to the main manuscript (Table 5). We address the impact of study quality in the Discussion (page 10, lines 243–246). Notably, the five studies with the highest QGenie scores reported no significant associations between individual SNPs and response to nCRT (page 10, lines 246–248), suggesting the need for cautious interpretation of positive findings from lower-quality studies.

Comments 4: The conclusion appropriately states that no SNP is ready for clinical use; however, a more explicit translational roadmap is needed.

Response 4: Thank you for this suggestion. We have expanded the Discussion (page 10–11, lines 267–283) to outline key requirements for future biomarker studies, including adequate sample size, standardized endpoints, multi-SNP predictive panels, and independent validation in external cohorts.

Comments 5: Tables summarizing SNPs could be improved by grouping variants by pathway, including allele frequencies, and adding effect directionality.

Response 5: Thank you for this constructive comment. We have revised Table 4 accordingly: SNPs are now grouped by biological pathway, allele frequencies have been added when reported, and effect directionality is indicated.

Comments 6: Typographical and stylistic corrections.

Response 6: We have revised the entire manuscript and made necessary stylistic and typographical corrections.

Comments 7: MGMT should be conceptually separated due to its involvement in epigenetic processes.

Response 7: We thank the reviewer for this clarification. We wish to emphasise that our review includes only germline SNPs, not somatic mutations or epigenetic alterations. For example, the KRAS variants analysed refer to the germline rs61764370 polymorphism in the 3′ UTR, rather than somatic codon 12/13 mutations. Similarly, the MGMT variants included are germline polymorphisms influencing baseline enzymatic activity, distinct from MGMT promoter methylation. We have clarified this distinction in the revised text.

Comments 8: The conclusion must be included in the manuscript.

Response 8: Thank you for noting this omission. A complete Conclusion section has now been added (page 11, lines 284–302).

Round 2

Reviewer 3 Report

Comments and Suggestions for Authors

The authors generally responded to the reviewers' comments and clarified the issues raised during session Question/Answer. In the original revised version of the manuscript, page and line numbers were incorrect, making it difficult to follow the authors' made changes. The revised version significantly improves the manuscript, but the page and line numbers must be correctly aligned.

Minor remark

The page and line numbers must be correctly aligned.

Comments on the Quality of English Language

Readable English.